# Learning Robust Real-World Dexterous Grasping Policies via Implicit Shape Augmentation

**Zoey Qiuyu Chen** [1]*  **Karl Van Wyk**[2]  **Yu-Wei Chao**[2]  **Wei Yang**[2]  **Arsalan Mousavian**[2]

**Abhishek Gupta**[1]  **Dieter Fox**[1,2]

[1]University of Washington    [2]NVIDIA
{qiuyuc, abhgupta, fox}@cs.washington.edu
{kvanwyk, ychao, weiy, amousavian}@nvidia.com

**https://sites.google.com/view/implicitaugmentation/home**

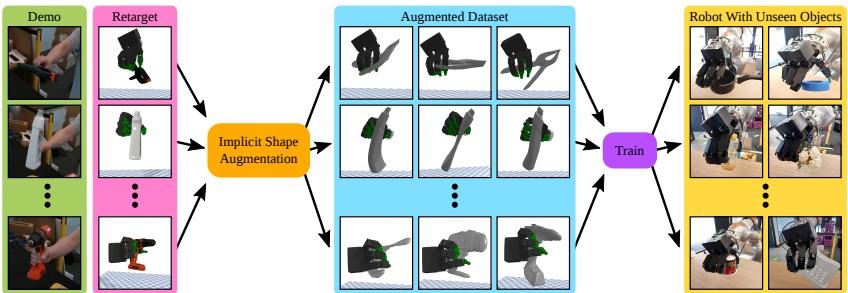

Figure 1: An illustration of the training scheme in ISAGrasp. A few human demonstrations are provided from motion capture. These demonstrations are retargeted to a four-fingered allegro robot in simulation. We use a correspondence-aware generative model to extrapolate the retargeted demonstrations to a large dataset of novel objects. We use this dataset to train a single grasping policy that is able to generalize to a variety of unseen objects in simulation and real world.

**Abstract:** Dexterous robotic hands have the capability to interact with a wide variety of household objects to perform tasks like grasping. However, learning robust real world grasping policies for arbitrary objects has proven challenging due to the difficulty of generating high quality training data. In this work, we propose a learning system (*ISAGrasp*) for leveraging a small number of human demonstrations to bootstrap the generation of a much larger dataset containing successful grasps on a variety of novel objects. Our key insight is to use a correspondence-aware implicit generative model to deform object meshes and demonstrated human grasps in order to generate a diverse dataset of novel objects and successful grasps for supervised learning, while maintaining semantic realism. We use this dataset to train a robust grasping policy in simulation which can be deployed in the real world. We demonstrate grasping performance with a four-fingered Allegro hand in both simulation and the real world, and show this method can handle entirely new semantic classes and achieve a 79% success rate on grasping unseen objects in the real world

**Keywords:** Dexterous Manipulation, Learning from Demonstration, Data Augmentation, Grasping

## 1 Introduction

Human hands are powerful tools for manipulating a wide range of objects. Our goal is to build dexterous robotic manipulators that can robustly and adeptly interact with human-centric environments. However, robustly controlling a dexterous hand remains challenging due to its high dimensional

---

*Work done while the author was a part-time intern at NVIDIA.

6th Conference on Robot Learning (CoRL 2022), Auckland, New Zealand.

state and action space and its multi-modal contact dynamics. Recent work has used deep reinforcement learning algorithms to learn complex tasks with dexterous manipulators [1, 2, 3, 4, 5, 6]. These methods often require careful reward engineering and environment design, and often lack generalization, struggling with robust real world deployment. An alternative paradigm involves collecting a large labelled dataset and using supervised learning (imitation learning). This has shown significant success with parallel jaw grippers [7, 8, 9] and suction cups [9, 10]. However, imitation learning methods have proven difficult to scale to multi-fingered robots due to the burden of human data collection. To scale robust policy learning to dexterous grasping, we require techniques that can leverage a small amount of human effort to learn robust, general grasping policies. This can be done by extrapolating a small number of human demonstrations to generate an abundance of data that interacts with *diverse* objects and is *successful* at grasping objects under real world dynamics. The key question is —how do we generate this type of diverse training data?

In this work, we propose *Implicit Shape Augmentated Grasping* (ISAGrasp), shown in Figure 1, a learning system that leverages a correspondence-aware generative model [11] to extrapolate a small number of human demonstrations to a large dataset of realistic objects and their corresponding grasps. In particular, we directly perform deformations on implicit 3-D shape representations of various objects in a learned latent space. The representation of point-wise shape deformations allows us to generate transformed grasps for novel objects that can be made successful with a small amount of active interaction in simulation. In doing so, implicit shape augmentation allows us to grow from a small dataset of human demonstration data for dexterous grasping to a significantly larger and more diverse dataset with novel object shapes and dynamically successful grasps. Given the dataset constructed by implicit shape augmentation, we can now perform large scale supervised learning. In this work, we represent a policy as predicting a pre-grasp and a final pose from pointclouds of the target object, with intermediate motion being performed with standard motion planning libraries. We show empirically that a policy learned via supervised learning on the dataset constructed by ISAGrasp is able to generalize widely across different objects in simulation and the real world. We demonstrate the efficacy of this pipeline on the rescaled YCB instances, ShapeNet and GoogleScans objects in simulation and achieve a 79% success rate on 22 unseen objects in the real world.

In summary, our contributions are (1) we propose a novel system, *ISAGrasp*, that is able to generate both a wide variety of objects *and* the corresponding dexterous grasps from a few human demonstrations. (2) We show that the augmented dataset can be used for learning point-cloud based control policies that are able to robustly grasp a large variety of *novel* objects in simulation (3) We demonstrate the efficacy of the proposed pipeline by deploying it in the real world. (4) We analyze the proposed pipeline through several ablation studies in simulation.

## 2   ISAGrasp: Dexterous Grasping Policies via Implicit Shape Augmentation

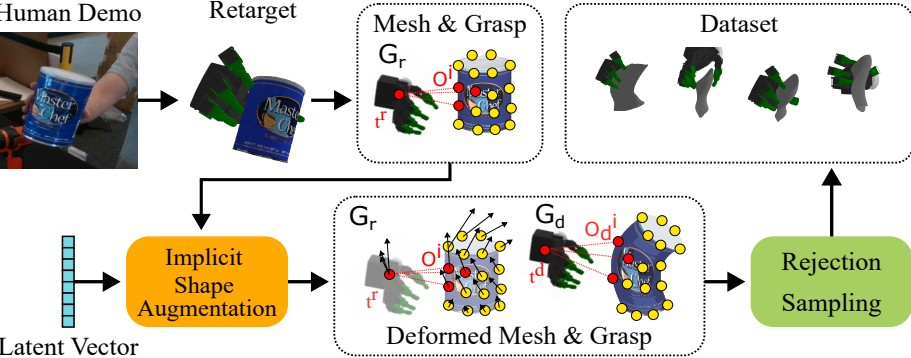

Figure 2: Implicit Shape Augmentation for generating augmented dataset from demonstrations. First a human demonstration is retargeted onto the Allegro hand to generate meshes and grasp labels. This data can then be used to generate a variety of new objects via shape augmentation with DIF-Net [11]. Grasps for these deformed objects can then be further refined with rejection sampling to generate dynamically consistent grasps.

To learn robust grasping policies that can operate in the real world for grasping novel objects of various shapes, we propose ISAGrasp —a framework for supervised learning of robust and general-

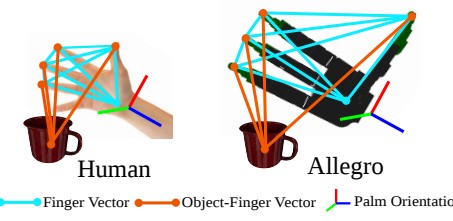

Human        Allegro

●—● Finger Vector    ●—● Object-Finger Vector    ⌐ Palm Orientation

$$d_g = \sum_{i=0}^{N} \|\vec{\mathbf{a}_{r_i}} - s_r \vec{\mathbf{a}_{h_i}}\|^2, \quad (1)$$

$$d_c = \sum_{i=0}^{N} \|\vec{\mathbf{c}_{r_i}} - \vec{\mathbf{c}_{h_i}}\|^2, \quad (2)$$

$$d_r = G(\mathbf{M}_r, \mathbf{M}_h), \quad (3)$$

$$\underset{(\mathbf{q}_r, \mathbf{f}_r, \mathbf{M}_r)}{\arg\min} \; (w_g d_g + w_c d_c + w_r d_r), \quad (4)$$

Figure 3: Illustration of retargeting a human hand demonstration to an Allegro hand

Figure 4: Optimization setup describing the retargeting problem from human to Allegro hand

izable grasping policies from successful grasps on a large variety of objects generated from a small set of human provided expert demonstrations.

Given a pointcloud of an object, we model a grasping policy as predicting (1) a pre-grasp pose $\mathbf{T}$ that controls robot hand's translation $\mathbf{T_t}$ and rotation $\mathbf{T_q}$, and (2) a final pose $\mathbf{G_f}$ that controls the 16-DoF finger pose that can firmly grab the object.

ISAGrasp assumes access to a set of labelled human grasping demonstrations, can be used to learn grasping policies via supervised learning. However, to learn truly robust and general grasping policies, this dataset must be grown to a much more diverse set of objects and successful grasps. Generating a multi-fingered grasping dataset of diverse yet realistic objects is a non-trivial problem [12, 13, 14]. We approach this problem by leveraging a correspondence-aware, deformation-based generative model to widely augment the set of human provided demonstrations, which can then be used to learn robust and general policies via supervised learning. An overview of our ISAGrasp system is shown in Figure 2, and we detail each component below.

## 2.1 Human-Robot Retargeting

ISAGrasp first collects $N$ human hand-arm demonstrations $\mathcal{D} = \{\tau_0^h, \tau_1^h, \ldots, \tau_N^h\}$ using motion capture, where each demonstration $\tau_i = \{(h_0^i, o_0^i), (h_1^i, o_1^i), \ldots, (h_T^i, o_T^i)\}$ is a trajectory of hand pose $h_t$ and object pose $o_t$. These demonstrations are then "retargeted" to a 22-DoF floating Allegro hand [15] in simulation. Note that this is a nontrivial problem since these demonstrations are in the morphology of the human hand and are typically not functional when directly mapped to a robot hand using standard Inverse Kinematics [16].

We formulate the retargeting objective as a non-linear optimization problem. First, to allow a robot to grasp in a similar pose to a human demonstration, we use the same cost function proposed in DexPilot [17], shown in Eq. 1, where $s_r$ is the ratio between the sizes of the robot and human hands, and each $\vec{\mathbf{a}_{r_i}}$ and $\vec{\mathbf{a}_{h_i}}$ is a displacement vector between finger tips and the palm of the robot and human hands, shown as blue in Figure 3. While Eq. 1 encourages grasp shape similarity, we additionally minimize the differences between the relative poses to the object, as captured by the vectors between fingers and the object center for the human grasp, $\vec{\mathbf{c}_{r_i}}$, and the retargeted grasp, $\vec{\mathbf{c}_{h_i}}$ (see Eq.2 and orange lines in Fig. 3). Finally, to orient the robot palm similar to the human hand, we add Eq. 3, which optimizes the minimum geodesic distance $G$ in SO(3) between the rotation matrix of the human palm $\mathbf{M}_h$ and the robot palm $\mathbf{M}_r$. The final retargeting goal is to find the 22 DoF configuration that minimizes the weighted sum of these three terms, which results in a candidate grasp $\mathbf{G_r}$ for each object in a human provided dataset.

As we describe in Section 2.3, the retargeted grasps are then refined via a rejection sampling step and used to generate the dataset with successful, dynamically consistent retargeted grasps. However, since this dataset is typically quite limited, we next outline how to augment the dataset with novel objects and grasps via implicit shape augmentation.

## 2.2 Implicit Shape Augmentation

Given a small set of object point clouds and successful retargeted grasps, $\mathbf{G_r}$, we build a large scale augmented dataset of novel objects and their corresponding successful grasps $\mathbf{G_d}$, using a correspondence-aware implicit generative model, DIF-Net [11].

To recap at a high level, DIF-Net aims to learn an implicit representation of 3D shapes as a scalar field. More specifically, DIF-Net represents a 3D shape via a instance agnostic template implicit field (which is common for all shapes of a particular category and represent the common features of a category), together with a latent conditional 3D deformation field (that perturbs this template field) which allows the shapes to be adapted to every particular object instance while maintaining semantic 3-D structure. Different novel but realistic shapes can be generated by sampling different latent vectors $\alpha$ and generating object deformations on known object classes using the learned deformation and correction fields, while maintaining semantics. This model naturally provides dense correspondences across object instances since different object instances are pointwise deformations on the same template.

We use this generative model in two ways: (1) leverage the ability to sample a variety of object instances by sampling latent vectors $\alpha$ to generate various novel objects via deformations and (2) the resulting dense correspondences allow us to estimate dynamically consistent grasps from human demonstrations to novel generated objects. Specifically, we use the latent conditional deformation field from a pretrained DIF-Net to generate novel instances and dynamically consistent grasps. To generate novel objects, we sample latent vectors $\alpha$ from a Gaussian that conditions a latent conditional deformation field. This deformation field generates point-wise deformations of particular object meshes chosen from the set of human demonstrations to generate novel objects. Second, to estimate the new grasps $\mathbf{G_d}$ for the deformed objects, we find N reference points on the original mesh that are close to the root position $t^r$ of the robot hand and compute the position $t_o$ of the robot relative to a local coordinate system $O_i$, centered at each reference point $i$: $t_i^o = O_i^{-1} \cdot t^r$. Once the object is deformed, we compute the corresponding coordinate system $O_i^d$ on the deformed meshes using the same deformation field, to estimate the new grasp location $t_d$ using the average of the local offsets: $t^d = \frac{1}{N}\Sigma_{i=1}^N (O_i^d \cdot t_i^o)$. In this way, the dense correspondences obtained via the DIF-Net are directly useful in generating grasps for novel, deformed object instances.

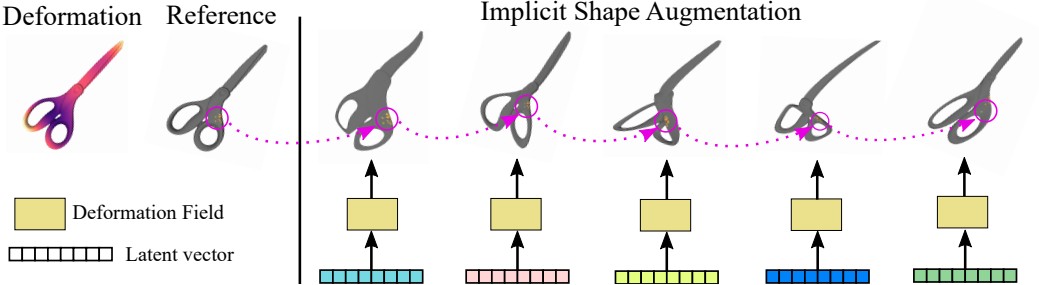

Figure 5: Deformation map and grasping correspondences for objects generated in ISAGrasp. Grasping correspondences on the original object (reference) and the deformed objects are highlighted inside the circle. As can be seen, object semantics are maintained. Different object instances are generated by sampling different latents

Figure 5 visualizes how objects and correspondences deform with different sampled latent vectors. We highlight the reference points on the original mesh, and their correspondences on deformed meshes (purple circles). The objects are deformed into a variety of realistic shapes while maintaining the original semantic structures and grasping correspondences.

Using the new grasp location $t^d$, together with retargeted orientation $q^r$ and finger pose $f^r$, we can construct new grasps $\mathbf{G_d}(t^d, q^r, f^r)$. While $\mathbf{G_d}$ are not guaranteed to be successful grasps, they provide good starting points for the local search procedure described next.

## 2.3  Grasp Refinement for Dynamics Consistency

Given transformed grasps $\mathbf{G_d}$, we use pose perturbation with rejection sampling to generate successful and dynamically consistent grasping poses. Since our shape augmentation model transforms successful grasps via the 3D deformation field, we found that only a small amount of perturbation is typically needed to find successful grasps for the deformed models using rejection sampling. In particular, we sample local perturbations $\delta_t$, $\delta_r$ and $\delta_f$ from a uniform distribution, and add these perturbations to the translation $t^d$, rotation $q^r$ and finger joints $f^r$ of the transformed poses $\mathbf{G_d}$. We evaluate success of each perturbed grasp $\mathbf{G_p}$ using a physics simulator (Pybullet [18]) and add domain randomization to save robust successful grasps and objects to the training dataset.

Using above methods, we can generate a large and successful dexterous grasping dataset with a variety of novel objects, and then perform supervised policy learning, as described in the next section.

## 2.4 Policy Learning via Supervised Learning

As described in the beginning of Section 2, we model the grasping problem as predicting a pre-grasp pose $\mathbf{T}$ and a final pose $\mathbf{G_f}$ given an object pointcloud observation. We perform a standard empirical risk minimization procedure on the above-mentioned dataset as an architecture and use a network consisting of PointNet++ SA modules [19] as a feature extractor. Instead of only feeding the raw point cloud to the network, we found significant improvements by appending object points with additional information regarding the alignment between the robot hand and the local object surface. In particular, we use the object point cloud $p$, the surface normal at each point $\vec{N}_o$, the normal

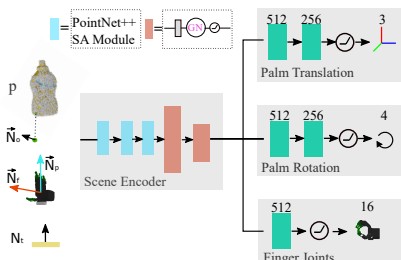

Figure 6: Policy network architecture. The point-net++ architecture inputs an object point cloud $p$, object surface normal $\vec{N}_o$, the table normal $\vec{N}_t$, robot facing direction $\vec{N}_f$ and the pointing direction $\vec{N}_p$ to generate palm translation, rotation and finger joints for the dexterous grasp.

of the table surface $\vec{N}_t$, and the hand facing direction $\vec{N}_f$ and pointing direction $\vec{N}_p$ to define the following feature vectors: $f(p)=(p_x, p_y, p_z, (\vec{N}_o \cdot \vec{N}_t), (\vec{N}_o \cdot \vec{N}_f), (\vec{N}_o \cdot \vec{N}_p), (\vec{N}_f \cdot \vec{N}_t))$. We append these vectors to each point in the point cloud. We found these features provide a compact description of alignment between the robot hand and the object, and using the relative vector alignments via the pairwise dot products allows us to improve the final grasping performance.

The network outputs a 3-dim translation and 4-dim quaternion, which are used to define the pregrasp pose $\mathbf{T}$. Additionally, the network predicts a 16-dim finger poses, which is used to define final pose $\mathbf{G}_f$. Figure 6 shows our network architecture details. We provide training details in Appendix A.

## 3 System Details

**Initial Dataset Construction.** We extracted human demonstrations from the DexYCB dataset [20], which contains mocap sequences $\mathcal{D}_{\text{human}}$ of humans hand poses $q_T^i$ picking up 20 YCB objects [21] with poses $o_T^i$. We picked 10 demonstrations per object, resulting in 200 demonstration in total. Appendix B 8.2 further explains our dataset choice.

**ISAGrasp Implementation Details.** We use a pretrained DIF-Net [11] to augment objects into a variety of novel shapes. First, we sample a 128-dim latent vector from a Gaussian distribution $\sim \mathcal{N}(\mu, \sigma^2)$, where $\mu = 0$ and $\sigma = 0.002$. Second, to estimate corresponding new grasps $\mathbf{G_d}$, we choose $N = 20$ closest points on the object surface as reference points and compute corresponding grasp location $t_d$ (Section 2.2). To obtain successful deformed grasps, we apply rejection sampling by uniformly sampling $\delta_t \in [-0.02\,m, 0.02\,m]$ for translation $t_d$, and $\delta_r \in [-0.5, 0.5]$ radians for rotation $q_r$, and apply the same perturbation $\delta_f \in [-0.1, 0.1]$ radians for finger joints.

**Hardware Setup.** We deploy our policy to a robotic platform that has 23 actuators across a KUKA LBR iiwa 7 R800 robot arm and a Wonik Robotics Allegro robotic hand, and use two cameras to provide necessary point cloud information. Details are provided in Appendix B.

## 4 Experiments

Through our experimental evaluation, we aim to answer the following questions:

1. Does ISAGrasp generate realistic novel objects based on a small set of scanned objects?
2. Does ISAGrasp allow for easy generation of dynamically consistent grasps on novel objects through grasp transformation and refinement?
3. Do policies learned on the dataset produced by ISAGrasp show improved robustness and generalization on unseen objects?
4. How does the training perform with different input features.

Table 1: Baselines evaluated in simulation

|  | RescaledYCB | ShapeNet | GoogleScans |
|---|---|---|---|
| Random | 0.03 | 0.05 | 0.02 |
| Train on successful random | 0.15 | 0.42 | 0.18 |
| Heuristic | 0.27 | 0.40 | 0.16 |
| Train on successful heuristic | 0.09 | 0.15 | 0.02 |
| Train on successful GraspIt | 0.29 | 0.42 | 0.20 |
| PPO with dense reward | 0.12 | 0.10 | 0.06 |
| DAPG +DR | 0.46 | 0.51 | 0.53 |
| DexYCB - DR - ISA | 0.34 | 0.35 | 0.22 |
| DexYCB + DR - ISA | 0.74 | 0.56 | 0.51 |
| **DexYCB +DR +ISA (ours)** | **0.74** | **0.74** | **0.70** |

We address these questions through a study in a PyBullet [18] simulation, followed by a real world experimental evaluation using the robotic system described in Section 3. We provide additional details of baselines in appendix B and more analysis on elements of the ISAGrasp in Appendix C.

**Evaluation Metrics.** We choose three unseen datasets with an increasing complexity: RescaledYCB, ShapeNet [22], and GoogleScans [23] (see appendix B). In particular, RescaledYCB contains 65 rescaled YCB objects, ShapeNet contains 200 unseen objects from "Can", "Bottle", "Mug" and "Bowl" categories, and GoogleScans objects contains 200 everyday objects. We place objects randomly on the table, and evaluate performance with 5 sets of object mass and friction. The success is defined as when the object is above the table by 10cm.

**Baselines.** Table 1 additionally compares the performance of our method to other baselines —**(1)** Randomly generate grasps around the object (Random) **(2)** use random baseline with rejection sampling to create successful dataset and train a policy **(3)** perform a predefined grasp where robot always grasps from the top (Heuristic). **(4)** train with Heuristic baseline with rejection sampling **(5)** train with Graspit!, an optimization-based grasp planner from prior work[24]. In addition, we train compare RL baselines: **(6)** one with a dense reward function using PPO[25] **(7)** One using an RL method (DAPG + DR) that combines an imitation learning objective via behavior cloning with reinforcement learning [26]. **(8)** train with domain randomization, but no shape augmentation (DexYCB + DR - ISA) **(9)** train with no shape augmentation or domain augmentation (DexYCB - DR - ISA) in order to understand the impact of shape augmentation and the impact of domain randomization. Please find further details of these baselines in the Appendix B.

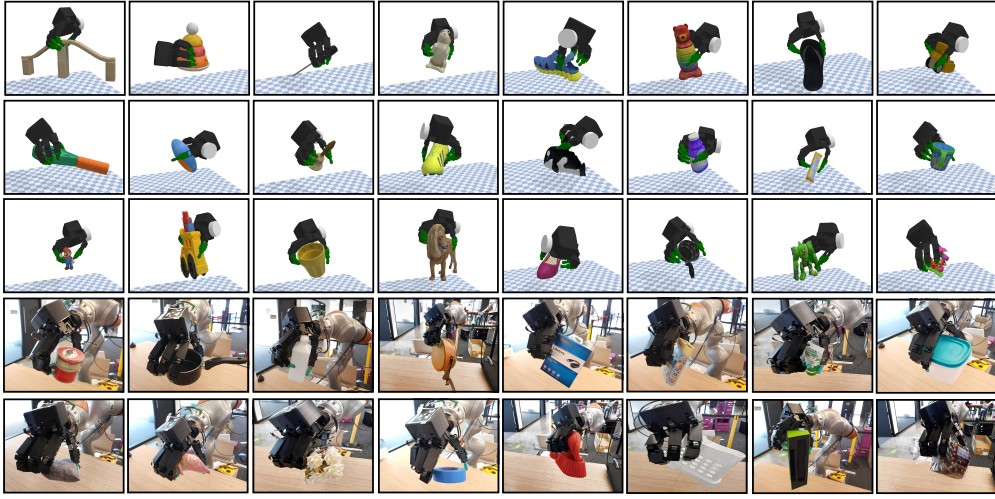

Figure 7: Qualitative results on unseen objects in simulation and in the real world. The top 3 rows shows the successful examples of our method on GoogleScans objects and the bottom 2 rows show successful examples using our policy deployed on unseen objects in real world.

Table 2: Real world test of ISAGrasp on 22 unseen objects. ISAGrasp is able to achieve 79% success rate

| obj | Tray | Crab | Chips | Box1 | Box2 | Bread | Flower | Pot | Roll | Hat | Honey |
|-----|------|------|-------|------|------|-------|--------|-----|------|-----|-------|
| #   | 3/5  | 4/5  | 4/5   | 2/5  | 4/5  | 5/5   | 3/5    | 4/5 | 4/5  | 5/5 | 5/5   |
| obj | Box5 | Scarf | Purse | Tape | Box3 | Bottle | Cream | Cap | Drill | Bag | Pringles |
| #   | 4/5  | 5/5  | 3/5   | 4/5  | 5/5  | 4/5   | 5/5    | 5/5 | 2/5  | 4/5 | 3/5   |

## 4.1 Simulation Results.

We first evaluate the efficacy of the ISAGrasp system on learning dexterous grasping policies in simulation. Table 1 shows the overall success rate of our method ISAGrasp on three unseen datasets in simulation. ISAGrasp (bottom row) is able to achieve 74% success rate on rescaled unseen YCB objects even the policy is only trained on augmented shapes. In addition, our method achieves 74% and 70% success rate on ShapeNet and GoogleScans objects. We describe analysis on how the various baselines perform in details in Appendix B.

## 4.2 Real World Experiments.

Next, we evaluated the grasping policy learned in simulation by ISAGrasp, directly in the real world. We perform directly simulation to reality transfer of the learned policy as described in Section 2.4. We evaluate our policy on 22 unseen real world daily objects (see Appendix Figure 17) and evaluate each object 5 times with random poses. We report number of success on each object in Table 2. On average, the policy is able to achieve 79% success rate on real world evaluation on novel objects. We observe several failure cases: (1) if the object is more transparent (Box1: container box), the pointclouds are incomplete and the network is less robust. (2) objects which require more careful grasping (Power drill) often have lower success rate. These show that we can leverage policies trained in simulation directly for real world grasping using ISAGrasp and that the robustness and generalization properties transfer from simulation to the real world.

## 4.3 Ablations and Analysis

We first provide some insights into the impact of various design decisions in ISAGrasp below:

**Impact of using correspondence-aware generative models:** To understand how important us-

| | Refinement Rate |
|---|---|
| ShapeGAN(R) | 0.09 |
| ShapeGAN(D) | 0.29 |
| DIF-Net(R) | 0.13 |
| DIF-Net(D) | 0.51 |
| **DIF-Net(Corr)** | **0.76** |

Figure 8: Analysis on shape generation. DIF-Net generates realistic and semantically meaningful objects (Left Panel) while ShapeGAN generates novel objects but often unrealistic and without any correspondences (Middle Panel). We show the refinement rate for obtaining successful grasps on these objects ( **R**: random grasps, **D**: retargeted demonstration, **Corr**: correspondence-guided grasps (right panel). The lack of correspondences makes refinement challenging, yielding only 30% success rate as compared to DIF-Net at 76%.

ing the dense correspondences provided by DIF-Net are for generating successful grasps on novel objects, we compare DIF-Net [11] with a correspondence-agnostic model, shapeGAN [14] by showing their generated meshes and the refinement rate for obtaining successful grasps. Refinement rate refers to the ratio of grasps that can be successfully refined via rejection sampling to the total number of proposed grasps. We use this metric to compare the efficiency of creating stable grasp datasets generated using different approaches. Shown in Figure 8 (Left), DIF-Net can smoothly deform the original shapes and generate more realistic shapes. We conduct 50 times of perturbation with rejection sampling on 100 objects generated by both methods. Figure8 (Right) shows refinement rate (**R**: random grasps without human demonstrations, **D**: using original human demonstrations without new grasps $G_d$ estimation, **Corr**: using new grasps $G_d$). Initialized with correspondence-guided grasps, DIF-Net (Corr) achieves 76% refinement rate while ShapeGAN(D) achieves below 30% with the same number of refinements. This indicates the importance of transferring grasps through a deformation based transformation for novel objects.

**Impact of Input Representation:** To understand the choice of input representation for supervised learning, we compare feature choices used as input for training policies, evaluated on objects from

the ShapeNet and GoogleScans datasets. We observe that using pointcloud $p$, surface normal vector $\vec{N_o}$, Robot vectors $\vec{N_f}$ and $\vec{N_p}$ performs best.

# 5   Related Work

Table 3: Ablations on input features

|  | $p$ | $p+\vec{N_o}$ | $p+\vec{N_o}+\vec{N_f}+\vec{N_p}$ |
|---|---|---|---|
| Success | 0.47 | 0.57 | 0.72 |

**Manipulation with Dexterous Hands.** To control a dexterous hand, prior work has investigated including planning with analytical models [27] and online trajectory optimization [28]. However, these methods assume accurate dynamics models and robust state estimates, which are difficult to obtain in complex real-world manipulation. Learning-based approaches particularly with deep reinforcement learning (RL) [29, 30, 5, 31, 32, 6, 2] have been investigated. Despite the progress, training deep RL models remains challenging due to high sample complexity and reward engineering. Although this issue has been mitigated by incorporating human demonstrations [26, 33, 3, 34, 1], these methods are still faced with a major challenge in scalability[3, 2]. Prior work has collected demonstrations kinesthetically [33], through VR interfaces [26], or using motion capture (mocap) solutions [35, 36, 37, 3], which are often limited in size. In comparison, the focus of our work allows policies to generalize to novel, unseen shapes, which could be easily combined with pipelines for improvement with RL with demonstrations [3, 1, 38, 26].

**Data Augmentation and Robustness.** Data augmentation has been used traditionally in vision tasks where images are cropped, rotated, normalized, etc [39, 40, 41, 42, 41, 42] to improve generalization and model robustness. In contrast, our work aims to generate novel objects with varying physical dynamics and does not simply learn invariant behavior but learns different grasping behavior for different objects. In a similar vein, domain randomization is used in robotics, where predefined parameters such as lighting, camera are randomized during training[43, 44] in order to learn a policy that is invariant to these parameters. However, this randomization does not aid with generalization to novel object shapes. More recently, methods have aimed to generate novel objects shapes programmatically [13], or use a generative model to create new objects in an adversarial setting [45]. This is. motivated similarly, but differs in being for parallel jaw grasping problems and not generating dynamically consistent grasping behavior via correspondences.

# 6   Limitations

**Challenging objects:** it's more challenging for our policy to succeed when the object is large or too flat. Since our shape augmentation is built on dexYCB dataset, it becomes more challenging if the test object is too different from the training objects, or requires a more specific way of grasping. The shape augmentation also does not cover objects of widely varying scales. See visualizations on challenging objects in Appendix D. **Real world experiments:** The method assumes access to a fairly complete point cloud. It will not succeed in scenarios with heavy amounts of occlusion or noise in the point clouds. **Functional grasping:** Currently the method does not do dexterous and functional grasps, and is only designed to lift the object. The utility of a dexterous hand is perhaps best utilized with functional grasps, but the current system does not optimize for this directly. **Accounting for kinematics:** The current system does not account for the kinematics of potentially hitting the table when the hand is mounted on a full arm setup. This should be accounted for as we build on this work in the future.

# 7   Conclusion

We present ISAGrasp, a novel system for learning dexterous grasping policies in the real world. ISAGrasp leverages a data augmentation approach that bootstraps a small number of human demonstrations with a large dataset with diverse and novel objects and grasps. By using a correspondence-aware generative model, we can deform original object shapes and generate dynamically consistent new grasps. We create a large and diverse grasping dataset and train a policy via supervised learning that can then be deployed in simulation and the real world on grasping novel objects, achieving over 75% success rate on grasping novel object instances in the real world.

## Acknowledgement

We thank Mohit Shridhar for providing helpful feedback on our initial draft. We thank Aaron Walsman for providing additional helps on figures and the draft. We are also grateful for Vikash Kumar, Aravind Rajeswaran, Jason Ma and Mandi Zhao for discussions on running RL baseline experiments. Part of this work was done while Qiuyu Chen was an Intern at NVIDIA. The work was also funded in part by an Intel gift.

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
