# OpenReview forum: "Learning Robust Real-World Dexterous Grasping Policies via Implicit Shape Augmentation"
_robot-learning.org/CoRL/2022/Conference — CoRL 2022 Poster_

### Official Review · Reviewer_RWRD · 2022-07-31

**Originality:** Very Good
**Technical Quality:** Good
**Clarity Of Presentation:** Good
**Impact:** 3

**Recommendation:**

Weak Accept: I recommend accepting the paper, but will not argue for my recommendation if the majority of other reviewers have a different opinion.

**Summary:**

This paper proposed a data-augmentation framework to learn dexterous grasp pose from point cloud observation. First, the DexYCB dataset which consists of human hand object interaction trajectories is used to provide an initial demonstration. Second, hand motion retargeting is utilized to convert the human hand motion to robot hand joint angles. Third, DIF-Net, an implicit neural network that can keep dense correspondence while deforming an implicit shape representation, is used to deform the original object mesh with sampled latent gaussian vectors. With correspondence after deformation, the grasp pose can also be modified according to the object. Then a rejection-sampling-based grasp refinement step is performed inside PyBullet simulator to eliminate physically-unfeasible grasp candidates. Finally, the author trained a PointNet++ to predict the palm pose and finger joints from point cloud input.

**Issues:**

- More details about the baselines
- More motivation on using the annotated dataset in the beginning
- Figure to illustrate the symbols in Section 2.2. It is better to give a better visualization on how to estimate a new grasp with symbols, e.g. t^d annotated on the figure.
- More clarification on comparison with the baseline method.


**Quality Of The Limitations Section:**

Limitations are addressed clearly

**Reviewer Expertise:**

4: The reviewer is confident but not absolutely certain that the evaluation is correct

**Robotics Focus:**

Sufficient demonstration on hardware

**Strengths And Weaknesses:**

This paper proposes a data augmentation strategy for dexterous grasping and motivates the problem well in the introduction. The proposed object-centric deformation method is very general and agnostic to tasks and manipulators. There are also several issues that need to be addressed in the paper as follows:

### Strength:

1. The proposed method is simple and elegant. It does not rely on dynamics analysis of the contact wrench space or other force-based measurements. The dynamics correctness checking is achieved solely by a physical simulator while the underlying troublesome computation is hidden below.
2. In principle, the implicit-based deformation network can generate infinite object mesh and corresponding grasp pose. This method is quite general.
3. The observation that “appending object points with additional information regarding the153
alignment between the robot hand and the local object surface” is beneficial for many robot-orientated regression tasks with point cloud input.


### Weakness:

1. The original demonstration dataset (DexYCB) seems not so useful in the whole pipeline, it only provided the initial object-grasp pair. The following data generated is achieved in a sample and reject fashion: sample with DIF-Net and reject with physical simulator. So the original dataset can be replaced by any method that can provide roughly okay grasp proposals, even if the grasp proposal itself is not dynamically-correct or successful. For example, the author can use any grasp proposal network, e.g. Contact Grasp Net, to generate a grasp pose and use the same data augmentation procedure for the following steps. In this case, the training data is not even limited to only YCB objects but any object meshes, which may lead to better generalization performance due to the increasing diversity of the training data. I assume that maybe one benefit of using human demonstration is for functional grasp when you do not want some grasp poses even if they are successful in simulator dynamically but not semantically feasible.

2. The PPO baselines with **sparse reward** seem too weak. For manipulation tasks with a dexterous hand, sparse reward will lead to nearly zero success rate. In this task, writing a distance-based dense reward is also very simple and straightforward, especially the authors can train the algorithm inside a PyBullet simulator.

3. The GraspIt baseline seems not used in a fair way: in the experiments, the author utilizes the results predicted by GraspIt and other heuristic to evaluate the grasp performance. However, the major contribution of this paper is the implicit shape-based augmentation, so a better way to compare GraspIt is as follows: i) replace DeYCB and implicit shape augmentation with GraspIt to generate diverse grasp poses ii) use PyBullet to reject sample the unfeasible grasp poses iii) train the same PointNet++ based on the data generated by GraspIt. Since the contribution is on the training data side, the GraspIt should also be used to generate training data but not evaluation. Otherwise the experiments may show the value of the Grasp Refinement for Dynamics Consistency but not Implicit Shape Augmentation.


**Summary Of Recommendation:**

Overall this is a good paper because it
1. focuses on important techniques,
2. are very general to tasks and manipulators, and
3. show large performance improvement in sim and real. In particular, the fact that the DIF-Net will also keep the semantic correspondence can extend the application of this paper to functional grasp, i.e. always hold the handle for a mug while deformation is used.

---

> ### Author Response · Authors · 2022-08-27
> **To Reviewer RWRD**
>
> Thank you for your feedback and suggestions! Please see our response to concerns below:
>
> 1. “More motivation on using DexYCB”
>
> > As you mentioned, humans are excellent at finding stable/feasible grasp based on years of experience, thus providing us a strong prior to generate good grasps much more efficiently. We add one experiment to compare the refinement rate of successful grasps during rejection sampling: under the same rejection sampling pipeline, if we initialize the grasp with GraspIt, only 26% grasps can be refined, while using humans as prior gives us 81% refinement rate.  In addition, DexYCB collects human grasps such that it’s easier to pass it to another person or do common tasks with that object. So learning from human demonstration provides a good prior for future work such as handover or human-robot collaboration tasks. We have added this to Section 4.4 of the paper.
>
> 2. “PPO with dense distance reward”
>
> > We have added a baseline (described as “PPO with dense reward” in the updated Table 1, Section 4) using PPO with a dense reward function defined as the negative distance between robot fingertips to the object. In addition, we added another contact reward to further encourage more fingers to contact the object. More formally, our reward function at each timestep is:
>
> >exp[-sum(||f_i -o||)] + 0.5*N_c/4
>
> >f_i refers to the position of finger tip i on the robot at each timestep, o is the object center, and N_c is the number of fingers contacting the object (Allegro has 4 fingers). This baseline improves over PPO with sparse reward as presented in the submission. We observe 30% success rate on training data, and 12% on rescaledYCB, 10% on shapenet and 6% on googlescan. While the success rate has improved, it is still significantly less successful than our proposed method due to augmentation and the stability of supervised learning.
>
> 3. “A better way to compare GraspIt is as follows…”
>
> > As suggested, we have added in a modification to our GraspIt baseline (described as “Train on successful GraspIt” in the updated Table 1, Section 4) as follows: 1. For each object in DexYCB, we first use GraspIt to generate 10 grasp candidates 2. We used the same rejection sampling with the same domain randomization to save stable grasps: at this stage, only 26% of grasps can be successfully refined. 3. We use the same network and train a policy using the same budget of time, and test on the same dataset: 29% on RescaledYCB, 42% on ShapeNet, and 20% on GoogleScan. Further descriptions of results and the method are in Appendix B, Section 2.2. We can see that using shape correspondences from ISAGrasp help getting significantly higher percentages of successfully refined grasps, showing the benefit of our method over directly using GraspIt.
>
> 4. “Figure to illustrate the symbols in Section 2.2”
>
> > We have added this in Fig 2, Section 2.
>
> 5. “More clarification on comparison with the baseline method/More details about the baselines”
>
> > We have added a detailed description in Appendix 9.2 of each of the baseline methods in Table 1, Section 4, along with several new baselines: PPO with dense distance reward, DAPG, train with successful GraspIt, DexYCB without domain randomizations. The baselines show the importance of both using augmentation to get robustness and generalization as well as refinement to get dynamically consistent trajectories.

---

> > ### Comment · Reviewer_RWRD · 2022-08-27
> > **Rsponse to Author Reply**
> >
> > Thanks for your great effort to update the new experiments and provides the response. The new version looks great to me and I do not have further concerns.

---

### Official Review · Reviewer_KGmi · 2022-07-31

**Originality:** Good
**Technical Quality:** Good
**Clarity Of Presentation:** Very Good
**Impact:** 3

**Recommendation:**

Weak Accept: I recommend accepting the paper, but will not argue for my recommendation if the majority of other reviewers have a different opinion.

**Summary:**

The paper proposes a system, Implicit Shape Augmentated Grasping (ISAGrasp), to augment limited human demonstrations of dexterous grasps.The implicit shape augmentation is built on DIF-Net [11], a correspondence-aware implicit generative model. Novel shapes are generated via deformation and the resulting dense correspondences helps transfer the human demonstration to novel objects. The transferred grasps are refined via simulation and a grasp prediction model is trained on the augmented dataset by supervised learning.

**Issues:**

* In Sec 4.3, refinement rate is not explicitly defined.
* Typos: L248, Figure8 -> Figure 8

**Quality Of The Limitations Section:**

Additional details required

**Reviewer Expertise:**

3: The reviewer is fairly confident that the evaluation is correct

**Robotics Focus:**

Sufficient demonstration on hardware

**Strengths And Weaknesses:**

Strengths:
* Human demonstration of dexterous grasping is expensive to collect and always require specialized set up. The paper proposes an interesting way to extrapolate the limited demonstrations to a large dataset of novel objects.
* The correspondence-aware generative model is used in a reasonable way, and the effectiveness of the method is shown by experiment.
* The system transfers well to real world and achieves a decent result.

Weakness:
* Analysis of the results is insufficient. The authors only show that "method A gets a better score than method B", but doesn't really explain why.
  * Why do the baselines, such as Heuristic and GraspIt perform so poorly? Are there any specific failure mode?
  * Why doesn't data augmentation improve the performance on RescaledYCB? Is it because the distribution of augmented objects are different from the YCB dataset, or it's something else?
  * I can understrand that augmentation can make the grasping policy generalize better to novel objects, but what are the source of remaining perfomance gap (~30% on 3 datasets)?
* The limitation part is not very satisfactory. I want to see more limitation on the algorithmic side, for example, what are the failure mode of the method and how can we further improve the success rate.

Questions:
* I like the second term (relative poses) of the regarting optimization. My question is why do you choose the center of the object as anchor point, instead of '''closest point on object surface'. Isn't the latter one more robust to shape variation and better modeling the contact?



**Summary Of Recommendation:**

Overall, I think it's an interesting paper the attempts to address the core problem of data-driven dexterous grasping - how scale up the dataset. Personally I'm not fully convinced that augmenting the dataset with generative model is the way-to-go, for example, the performance on YCB-rescaled is not improved after data augmentation. But it could still be a contribution to the community, given that the authors can provide more interpretations of results.

---

> ### Author Response · Authors · 2022-08-27
> **To Reviewer KGmi Part 1**
>
> Thank you for the insightful feedback! We address individual concerns below:
>
> 1. “Why do the baselines perform so poorly? Are there any specific failure modes?”
>
> > We add descriptions of this in Section 4.1 and visualize baseline failure mode in Appendix Figure 14.
>
> > (1) Heuristics baseline: We use a heuristic that we have seen being used for grasping in practical systems: grasp the object from the top, with a fixed 5cm offset from the object top surface. This method usually works when the object is small and round (can, small box), but less likely to succeed when (a) the object is less symmetric like “a mug with a handle” (Appendix Figure 10 top row) and requires more careful reorientation of the pregrasp pose or (b) the object is unstable to grasp from the top (Appendix Figure 10 middle row).
>
> >(2) GraspIt: GraspIt is based on optimization using the contact energy function, but does not account for dynamics, or ensure stability.
> Failure modes include:
> (a) Collision checking: Starting from an open palm, we interpolate finger poses into the final graspIt pose. However, graspIt often fails when the robot hand is in collision with a table or starts changing the object pose while closing the fingers (Appendix Figure 14 Bottom a).
> (b) Unstable grasps: GraspIt doesn’t take dynamics into consideration, making the produced grasps potentially unstable while lifting up the object (Appendix Figure 14 Bottom b&c).  In contrast, our rejection sampling with domain randomization encourages more stable grasps. (c) Challenging to optimize: When the object surface is more complex (Appendix Figure 14 Bottom d), it is usually more challenging for GraspIt to find a good grasp solution.
>
> >(3) Random baseline: Random is referring to generating a robot pose randomly around the object. Because the action space of an allegro hand is high-dimensional, (22DoF for floating hand, 23Dof for Kuka-Allegro), randomly generating a stable grasp is much less likely and almost fails on every object.
>
> 2. “Why doesn't data augmentation improve the performance on RescaledYCB?”
>
> > The rescaled YCB dataset is created by scaling different dimensions of objects in the DexYCB dataset (as shown in Appendix Figure 16). This results in objects of widely varying sizes, rather than very different shapes. Since shape augmentation via ISAGrasp largely provides robustness to shape, and less prominently to scale, it is unable to effectively generalize to objects in rescaled YCB. We verify this quantitatively by performing experiments where we rescale all dimensions of the RescaledYCB dataset such that the object is roughly back to original size (although some dimensions may be more stretched than others). We observe 81% with shape augmentation and 75% without augmentation on this new dataset. This further shows that inability to generalize to different scales is the limiting factor here. Supplementing ISAGrasp with large dimension scales would further help.

---

> > ### Author Response · Authors · 2022-08-27
> > **To Reviewer KGmi Part 2**
> >
> > 3. “What is the source of the remaining performance gap?”
> >
> > >We add a description in Section 4.1
> >
> > > The remaining performance gap stems from several factors:
> >
> > >(1) Generalization error: We train on augmented dexYCB objects but test on rescaledYCB, ShapeNet and GoogleScan datasets. Despite the shape augmentation, there is some distribution shift between the train and test datasets, which results in some amount of the gap in performance. This is verified by seeing that the performance on the same training objects is 81%, while unseen objects on average is 73%.
> >
> > >(2) Compounding error: The second cause of error is that we subsample the point cloud into 1024 points for GPU memory reasons. This may lead to some loss of performance since some shape properties could be lost.
> >
> > >(3) Incomplete coverage: In order to cover unseen object poses, during training, we do domain randomization on object poses by adding orientation perturbation to the original object pose and the corresponding pregrasps. However, this does not guarantee all unseen poses are included during training.
> >
> > >The performance of both our method and the baselines are best appreciated from the videos on our supplementary website and through the analysis figures added in to Appendix Fig 14 to Figure 16.
> >
> > 4. “The limitation part is not very satisfactory”
> >
> > > Thank you for bringing this to our attention. We will update with additional limitations below:
> >
> > > (1) Poor performance on challenging objects: it’s more challenging for our policy to succeed when the object is large (e.g. cracker box lying on the table Appendix Figure 15a) or too flat (scissors in Appendix Figure 15b)).  Since our shape augmentation is built on dexYCB dataset, if the test object is too different from the training objects, or requires a more specific way of grasping (e.g. Eagle with two wings: Appendix Figure 15c). The shape augmentation also does not cover objects of widely varying scales.
> >
> > >(2) Real world experiments: The method assumes access to a fairly complete point cloud. It will not succeed in scenarios with heavy amounts of occlusion or noise in the point clouds.
> >
> > >(3) Functional grasping: Currently the method does not do dexterous and functional grasps, and is only designed to lift the object. The utility of a dexterous hand is perhaps best utilized with functional grasps, but the current system does not optimize for this directly.
> >
> > >(4) Accounting for kinematics: The current system does not account for the kinematics of potentially hitting the table when the hand is mounted on a full arm setup. This should be accounted for as we build on this work in the future.
> > We have added these to Section 6.
> >
> > 5. “why do you choose the center of the object as anchor point, instead of closest point on object surface”
> >
> > > We need to find a fixed reference point from the object in order to perform retargeting. While more generally, any fixed reference point would also work (including closest point on object surface), we choose object center instead because it’s more stable and doesn’t vary over time as the hand is moving close to the object.
> >
> > 6. “Refinement rate is not explicitly defined”
> >
> > > Refinement rate refers to the ratio of grasps that can be successfully refined via rejection sampling to the total number of proposed grasps. We use this metric to compare the efficiency of creating stable grasp datasets generated using different approaches.

---

### Official Review · Reviewer_kqPQ · 2022-07-31

**Originality:** Very Good
**Technical Quality:** Good
**Clarity Of Presentation:** Very Good
**Impact:** 4

**Recommendation:**

Weak Accept: I recommend accepting the paper, but will not argue for my recommendation if the majority of other reviewers have a different opinion.

**Summary:**

This paper studies the problem of dexterous grasping. It proposes to use correspondence-aware implicit deformation networks to propagate a small number of human grasp demonstration to grasping configurations on a diverse deformed object set. Trained with the generated objects and corresponding grasp configurations, the grasping policy can better generalize to unseen objects in human demonstration.

**Issues:**

Fix the comparison with the baseline:
Collect grasping data with rejection sampling on (augmented) shapes using Random, Heuristic and GraspIt!. Use a same budget as the proposed method and compare rejection rate and the final performance of same model trained on these different data.

**Quality Of The Limitations Section:**

Limitations are addressed clearly

**Reviewer Expertise:**

3: The reviewer is fairly confident that the evaluation is correct

**Robotics Focus:**

Sufficient demonstration on hardware

**Strengths And Weaknesses:**

Strengths:
1. It's a smart idea to augment human grasping demonstrations with correspondence-aware shape deformation networks. It simultaneously creates novel shapes and the corresponding grasp configurations.
2. Simulation experiments demonstrate the efficacy of the proposed shape augmentation in the comparison between last two rows in Table 1.
3. This paper conducted real-world experiments of dexterous grasping, which makes the results much stronger.

Weaknesses:
1. Table 1: The comparison with baselines seems not very fair. Random, Heuristic and GraspIt! should be compared with transformed grasps Gd when performing the rejection sampling (133). Collect data with rejection sampling on augmented shapes using Random, Heuristic and GraspIt! with a same budget and compare the result.
2. 248: what does refinement rate mean?

Minors:
220: baseelines -> baselines
Fig 8 caption: miss one '(' before "Middle Panel)", near the end of the second line

**Summary Of Recommendation:**

The idea of this paper is very novel and interesting, but the comparison with baselines is not very fair so it's hard to tell how much gain does the proposed method bring. I believe this can be fixed so I vote for weak accept.

---

> ### Author Response · Authors · 2022-08-27
> **To reviewer kqPQ's concerns:**
>
> Thank you for your suggestions and feedback! Please find detailed responses to concerns below:
>
> 1. “ The comparison with baselines seems not very fair…collect data with rejection sampling on augmented shapes using Random, Heuristic and GraspIt! with the same budget and compare the result.”
>
> > As suggested, we tried collecting data with the same rejection sampling on random/heuristic/graspit baselines on shape-augmented objects and found that they yielded 9%, 38%, 21 % successful grasps as compared to 76% for ISAGrasp. This is because ISAGrasp is able to use correspondence awareness to transform the initial grasp accordingly so grasp refinement improves. Due to computational budget constraints, the complete training pipeline with supervised learning was not able to be run till completion during the rebuttal period, but given the success rate of refinement during dataset generation, the results are likely to have the same trend. We will include this in the final version. We also added three experiments with non-augmented shapes (the original DexYCB objects) to further provide further ablations. We use random, heuristic and GraspIt to generate 10 candidate grasps for each dexYCB object, apply the same amount of rejection sampling with domain randomizations. At this stage, only 26%, 30% and 26% are successfully refined. We train policies using supervised learning on these successful grasps and observe performance of 25% (Random), 9% (Heuristic), and 29% (GraspIt) on average on the three unseen datasets. While our approach is able to achieve 73% on average.
>
> 2. “What does refinement rate mean?”
>
> > Refinement rate refers to the ratio of grasps that can be successfully refined via rejection sampling to the total number of proposed grasps. We use this metric to compare the efficiency of creating stable grasp datasets generated using different approaches.  We have clarified this in the paper.
>
> 3. “Typos:”
>
> > Thank you for the pointers. We have updated the pdf accordingly.

---

### Official Review · Reviewer_gvHJ · 2022-08-03

**Originality:** Very Good
**Technical Quality:** Good
**Clarity Of Presentation:** Very Good
**Impact:** 3

**Recommendation:**

Weak Accept: I recommend accepting the paper, but will not argue for my recommendation if the majority of other reviewers have a different opinion.

**Summary:**

The authors present ISAGrasp, a method for performing dexterous grasps on arbitrary objects using a pre-trained network from a large dataset augmented by a generative model which outputs corresponding grasp points on deformed meshes. This is achieved with a combination of labeled human grasp demonstrations with pose-retargeting, to get corresponding robot (Allegro) grasps. To generate novel objects and grasps, they use DIF-Net, building upon prior work by Deng et al. '21, which returns point-wise deformations of the objects used in the human demonstrations. They compute new grasp locations that minimize the total offset to the original grasp over a local patch of points. Experimentally, their method outperforms RL, heuristic, and grasping baselines.

**Issues:**

Summarizing the issues described in the summary of recommendation:

1.  The RL algorithm used (PPO w/ and w/out demonstration) does not make a fair/useful comparison as this method is not designed to work with demonstrations.
2. There is no direct comparison between shape augmentation and other forms of domain randomization (such as initial grasp/object pose/force perturbations), which could be used when learning a grasping policy instead.
3. In addition to a better RL method to compare to (such as DAPG or simple Dagger/supervised learning), the paper should further highlight the advantages/differences to similar existing methods involving RL + learning from demonstration (such as DexMV and DexVIP). It is unclear whether these methods would perform better or worse than the grasping policy learned with shape augmentation given the same demonstration dataset/retargeting inputs. Mentioning the trade-offs/similarities to these approaches is not clearly discussed in the Related Work section (lines 269-270), and should be expanded further.

I have updated my original review as these issues have been addressed by the authors in their rebuttal.

**Quality Of The Limitations Section:**

Additional details required

**Reviewer Expertise:**

4: The reviewer is confident but not absolutely certain that the evaluation is correct

**Robotics Focus:**

Sufficient demonstration on hardware

**Strengths And Weaknesses:**

Strengths
- Their grasping policy generalizes to novel object instances, provided that an accurate point cloud of the object can be generated from the scene.
- Their method extensively compares to different grasping approaches with a wide range of objects, in addition to ablating the input features used by the grasping policy. These comparisons show strong benefits to shape augmentation.

Weaknesses
- By using an open-loop policy, their method is unable to react to dynamic/cluttered environments or recover from failed grasps
- Their simulation does not account for kinematic infeasibility with the environment, both for the arm and the hand.
- This method does not consider functional/dexterous grasps of objects, and therefore does not leverage the dexterity of the hand.
- Their method uses demonstrations, but does not leverage an RL approach which is designed for use with demonstrations

**Summary Of Recommendation:**

Overall, this method is novel in the scope of works that have looked at dexterous grasping, as shape augmentation to scale human demonstrations has not been previously studied. For this, I think the paper does a convincing job of laying out the approach and comparing to other possible approaches.

My only issues with the paper in its current state are that a) it does not compare to any approaches that also leverage learning from demonstration (which is perhaps the most direct comparison), and b) the RL method used is not well suited for use with demonstrations. As Table 1 does not really show RL being very successful at this task, it would make more sense to instead compare to a method such as DAPG (Rajeswaran et al. '18) to leverage such demonstrations. As for methods that involve learning from demonstrations, both DexVIP (Mandikal and Grauman '21) and DexMV (Qin et al. '21) would be effective comparisons. Another option could also be to compare to GAIL or offline RL approaches such as TD3 or SAC (in simulation). As both of these works also consider pose-retargeting and similar robot grippers, it would be worth trying to see if shape/grasp augmentation can benefit these methods as well (as this is the core contribution of this work).

As for the datasets used, the authors do leverage an extensive set of objects from the YCB, ShapeNet, and GoogleScans datasets, and do show a significant difference in performance on these objects when compared to the baselines. One further comparison to aid in showing what advantage shape augmentation has over domain randomization would be to compare to another grasping policy that is only trained with standard domain randomization (adding perturbations to the initial grasp/object pose/forces on the object).  This would help isolate how much shape augmentation improves the policy compared to each of these randomizations (and when used in conjunction with).

---

> ### Author Response · Authors · 2022-08-27
> **To reviewer gvHJ's concerns**
>
> Thank you for your thoughtful feedback! Please see our response to each concern below.
>
> 1. “The RL algorithm used (PPO w/ and w/out demonstration) does not make a fair/useful comparison as this method is not designed to work with demonstrations.”
>
> > To provide more fair comparisons to baseline methods, we added DAPG [1] based on its original code and parameters as suggested. We replaced the original MLP policy with our pointnet++ based network to deal with point cloud inputs. We trained DAPG for ~3 days and evaluated it on three unseen datasets: RescaledYCB, Shapenet and Googlescan, obtaining success rates of 46%, 51% and 53% vs the 74%, 74% and 70% achieved by our method. This inferior performance is likely due to optimization challenges in multi-task RL learning as shown in prior work [2, 3]. Additionally, because it lacks shape augmentation, DAPG does not generalize widely.
> Moreover, we added one more RL baseline and trained PPO with a dense distance reward, which only achieves 12% (RescaledYCB), 10% (ShapeNet) and 6% (GoogleScan) success rate, which is significantly worse than our method. Please see our updated results in Table 1.
>
> 2. “There is no direct comparison between shape augmentation and other forms of domain randomization (such as initial grasp/object pose/force perturbations):”
>
> > In the submitted draft of the paper, we did indeed use domain randomization when generating the grasp dataset for training. We randomized object mass, friction, and force perturbation to filter unstable grasps during rejection sampling (see domain randomization details in Section 4 of the updated pdf). In the original Table 1, the entry labeled “DexYCB without augmentation” shows the success rate using these domain randomizations, but without any shape augmentation (this label has been updated in the revised draft as DexYCB +DR -SA). We see that shape augmentation provides close to 20% improvement on the unseen objects: ShapeNet and GoogleScan.
> >
> >To highlight the impact of such domain randomization, we added an experiment (labeled “DexYCB -DR -SA” in the updated Table 1) to compare training on DexYCB without any of these domain randomizations, and observe 34% performance rescaledYCB, 35% performance on shapenet, and 22% performance on shapenet, which is a significant drop from the “DexYCB +DR -SA” baseline which uses domain randomization.
>
> 3. “the paper should further highlight the advantages/differences to similar existing methods involving RL + learning from demonstration”
>
> > We have added a discussion of this to Section 5 of the updated paper above. While our work uses human examples to seed the initial pre-grasp positions, the key contributions of our work are complementary to works like DexMV [4] and DexVIP [5] (learning from videos), and methods like DAPG [1] and DDPGfD [6] (Learning from demonstrations with RL). DexMV and DexVIP aim to extract hand pose from video and use this as prior for training in simulation, but do not train on augmented objects for generalization. Similarly, DAPG and DDPGfD initialize from demonstrations and improve with RL on a fixed set of known objects. In comparison, the focus of our work is on learning grasping policies that generalize by explicitly generating novel, augmented shapes and then training robust policies on these shapes. This allows policies to generalize to novel, unseen shapes. Our shape augmentation method ISAGrasp could be easily combined with pipelines for improvement with RL with demonstrations (such as DAPG/DDPGfD) or for imposing priors from human video  (such as DexMV/DexVIP).
>
> [1] Rajeswaran, Aravind, et al. "Learning complex dexterous manipulation with deep reinforcement learning and demonstrations., RSS, 2017
>
> [2] Yu, Tianhe, et al. "Gradient surgery for multi-task learning.", NeurIPS, 2020
>
> [3] Liu, Bo, et al. "Conflict-averse gradient descent for multi-task learning." NeurIPS, 2021
>
> [4] Qin, Yuzhe, et al. "Dexmv: Imitation learning for dexterous manipulation from human videos.", ECCV, 2022
>
> [5] Mandikal, Priyanka, and Kristen Grauman. "Dexvip: Learning dexterous grasping with human hand pose priors from video." CoRL, 2021
>
> [6] Vecerik, Mel, et al. "Leveraging demonstrations for deep reinforcement learning on robotics problems with sparse rewards." arXiv,  2017.

---

### Meta-Review · Area_Chair_yRps · 2022-08-10

**Recommendation:** Accept (Poster)
**Confidence:** 5

**Metareview:**

All the reviewers acknowledge somehow the novelty/originality of the paper but have all questioned the proposed approach in terms of baselines and fair comparisons, e.g.,  PPO, GraspIt. In the phasse of rebuttal, the authors have significantly reinforced the comparison of some well-known baselines suggested by the reviewers. As a result, all the reviewers achieved a consensus for weak accept

**Best Paper Nomination:**

No

---

> ### Author Response · Authors · 2022-08-27
> **General Response**
>
> We apologize for the delayed response. Due to reviewers’ request for additional baselines, we wanted to prepare these experiments before making a full response. We thank all reviewers for their constructive feedback! To address the concerns raised, we performed several experiments which we describe below, followed by individual responses to reviewer concerns.
>
> 1. PPO baseline: As suggested by reviewer RWRD, we modified sparse reward PPO with a dense distance based reward function as a baseline. This baseline performed better than the version reported in the paper, showing  30% success rate on training set, 12% on RescaledYCB, 10% on ShapeNet, and 6% on GoogleScan. Despite these improvements, this is still significantly lower than our approach (74% RescaledYCB, 74% Shapenet, 70% GoogleScans). The results have been updated in Table 1.
>
> 2. DAPG baseline: As suggested by reviewer gvHJ, we ran a comparison with the DAPG algorithm proposed in [1], which combines an imitation learning objective with an RL objective. We modified this to work with point clouds by using a pointnet++ style architecture.  We found that DAPG indeed improves the pure RL without learning from demonstration. While DAPG can be quite successful for a single objective (showing 83 % success), it struggles in the multi-object case, only achieving 46% on RescaledYCB, 51% on ShapeNet, and 53% on GoogleScan (likely because of the optimization challenges of multi-task RL [2, 3]), significantly lower than our approach. The results have been updated in Table 1.
>
> 3. GraspIt baseline: As suggested by reviewer RWRD , we modified our GraspIt baseline by doing the following: (1) For each object in DexYCB, we first use GraspIt to generate 10 grasp candidates. (2) We used the same rejection sampling to save stable grasps: at this stage, only 26% of grasps were successfully refined. (3) We train a policy with supervised learning. We find that the success is 29% on the RescaledYCB dataset, 42% on Shapenet, and 20% on GoogleScan as compared with 74%, 74%, and 70% for our method. The results have been updated in Table 1.
>
> 4. Random/heuristic/graspit baselines with rejection sampling: As suggested by reviewer kqPQ , we augment the random/heuristic/graspit baselines with rejection sampling to reject infeasible grasps. This allows these methods to obtain 9%, 38%, 21% refinement rates respectively when generating the grasp dataset, which is still significantly lower than the 76% obtained with ISAGrasp.
>
> 5. Domain Randomization: Reviewer gvHJ suggested that we perform other types of domain randomization beyond shape randomization. In fact we already do so in the experiments in the paper, the “DexYCB without augmentation” (shown in the original pdf, table 1) baseline is performed with weight, force, friction and initial pose perturbations, but without shape augmentation. To analyze the importance of these randomizations, we have also added an experiment without any of this domain randomization, which is updated in Table 1 (labeled “DexYCB -DR -SA“). We find that this achieves a 34% success rate on RescaledYCB, 35% on Shapenet, and 22% on googleScan. This is a significant degradation over both domain randomization and shape augmentation (“DexYCB +DR +SA (ours)”: 74%, 74%, 70%).
>
> 6. Analysis: We performed a deeper analysis of the experimental results. In particular, we provide examples of failure modes of the various baselines in Appendix Figure 14. We analyze the differences between the different datasets tested in Appendix Figure 16 and we show the various sources of error in Appendix Figure 15. A description is in Section 4.1 and Section 4.4. We have updated the PDF below.
>
> Please find responses to individual concerns below.
>
> [1] Rajeswaran, Aravind, et al. "Learning complex dexterous manipulation with deep reinforcement learning and demonstrations., RSS, 2017
>
> [2] Yu, Tianhe, et al. "Gradient surgery for multi-task learning.", NeurIPS, 2020
>
> [3] Liu, Bo, et al. "Conflict-averse gradient descent for multi-task learning." NeurIPS, 2021

---

> > ### Author Response · Authors · 2022-08-27
> > **updated draft with Appendix**
> >
> > Here is the updated pdf with appendix.